# Light-driven radical copper-catalyzed allylic amination via allylic copper intermediates

Hang Luo[1,2,6], Yupeng Yang[3,6], Yunpeng Ma[1,6], Fangnian Yu[1], Min Gao [4]✉, Shizhao Xu[1], Huawei Ju[5], Yang Li [5], Kaifeng Wu [3]✉ & Luqing Lin [1,2]✉

The synthesis of allylic amines through an outer-sphere nucleophilic substitution mechanism involving electrophilic allyl copper(III) complexes with soft amines represents an uncharted territory in catalysis. This study introduces a radical-based approach for the generation of allylic copper(III) complexes, enabling the efficient synthesis of allylic amines in the presence of alkyl/aryl amines. Through copper photocatalysis, we demonstrate a radical-induced small-ring opening process that produces allylic amines featuring skipped double bonds, while simultaneously achieving highly regioselective 1,4-carboamination of 1,3-dienes with high *E/Z* selectivity. Mechanism studies substantiate the radical-mediated formation of allylic copper complexes and provide evidence for the involvement of outer-sphere nucleophilic substitution at allylic copper(III) complexes.

Allyl amines constitute a pivotal structural motif ubiquitously present in pharmaceuticals, agrochemicals, and natural products, while also serving as versatile building blocks in synthetic organic chemistry[1–4]. Among established synthetic methodologies, transition-metal-catalyzed allylic substitution reactions, particularly Tsuji-Trost reactions, have emerged as a robust platform for constructing diverse allyl amines through π-allyl metal intermediates that mediate carbon-nitrogen bond formation[5,6]. This catalytic approach addresses a key limitation of traditional allylation methods using allyl (pseudo)halides, wherein uncatalyzed conditions often lead to uncontrolled over-allylation when employing simple alkyl/aryl amines as nucleophiles. Noble-metal based systems, especially palladium system, dominate current Tsuji-Trost methodologies (Fig. 1a)[7–10]. While copper-catalyzed processes involving Cu[II/I] redox cycles have been proposed as mechanistically distinct alternatives to the conventional two-electron Pd[II/I] catalytic paradigm[11–13], the direct synthesis of allylic amines via nucleophilic substitution at allylic Cu(III) complexes remains unexplored[14], highlighting a critical research gap in the application of copper catalysis to this fundamental transformation.

Copper catalysis has emerged as one of the most powerful methods for facilitating C-C and C-heteroatom cross-coupling reactions[15–19]. High-valent organocopper complexes are often proposed as intermediates in these metal-catalyzed processes, although they are rarely isolated[11–13,20]. The proposed Cu(III) intermediates exhibit high reactivity but typically elude detection. Consequently, they play a crucial role in the formation of C-C and C-heteroatom bonds during the final step, which involves the release of coupling products via inner-sphere reductive elimination of copper(III) intermediates (Fig. 1b). The oxidation addition of the C(sp3)-X bond to a Cu(I) species is generally regarded as the rate-limiting step in copper-catalyzed cross-coupling reactions of alkyl electrophiles. Such reactions often necessitate the pre-coordination of a deprotonated-nucleophile (Nu⁻) to form a Cu(I)-Nu complex before C(sp³)-X bond activation can occur. For example, recently, several significant advances have been made in copper catalysis to activate C(sp³)-X bond for alkynylation[21,22], arylation[23–25], amination[26,27], etherification[28]. Similarly, in the allylation process, the generation of allylation products proceeds through an inner-sphere reductive elimination step involving

¹Central Hospital of Dalian University of Technology, School of Chemistry, Dalian University of Technology, Dalian, Liaoning, China. ²State Key Laboratory of Elemento-Organic Chemistry, Nankai University, Tianjin, China. ³State Key Laboratory of Chemical Reaction Dynamics and Collaborative Innovation Center of Chemistry for Energy Materials (iChEM), Dalian Institute of Chemical Physics, Chinese Academy of Sciences, Dalian, Liaoning, China. ⁴Institute for Chemical Reaction Design and Discovery (ICReDD), Hokkaido University, Sapporo, Hokkaido, Japan. ⁵State Key Laboratory of Fine Chemicals, Dalian University of Technology, Liaoning, China. ⁶These authors contributed equally: Hang Luo, Yupeng Yang, Yunpeng Ma. ✉e-mail: gaomin@icredd.hokudai.ac.jp; kwu@dicp.ac.cn; linluqing@dlut.edu.cn

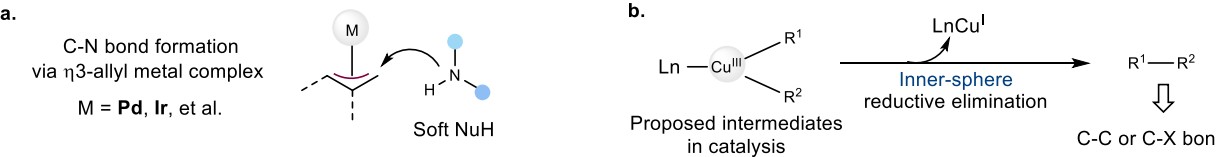

**c. Strategies to form C-N bond via nucleophilic substitution to electrophilic allylic copper complexes**

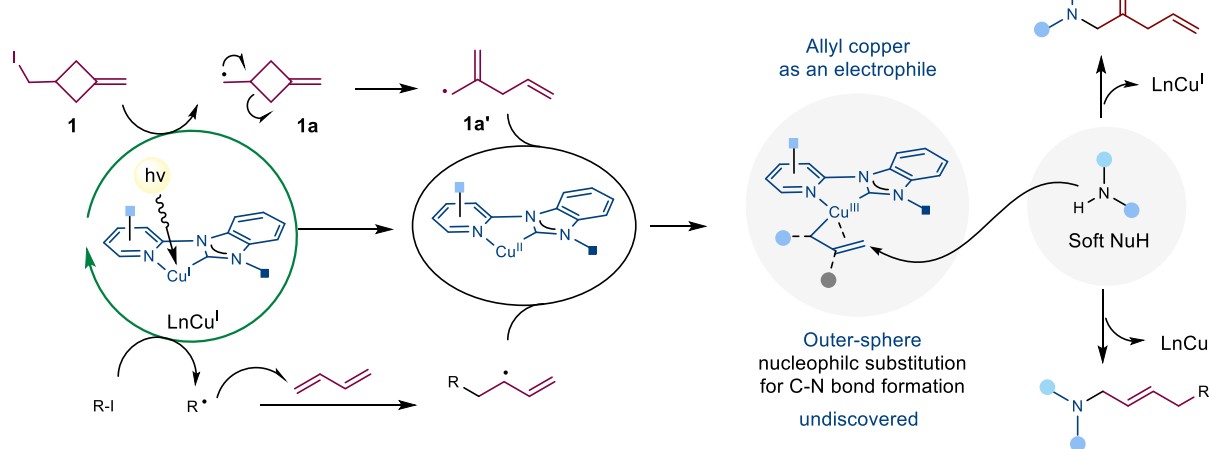

**d. Optimization of conditions for synthesis of allylic amines via two or three-component reactions**

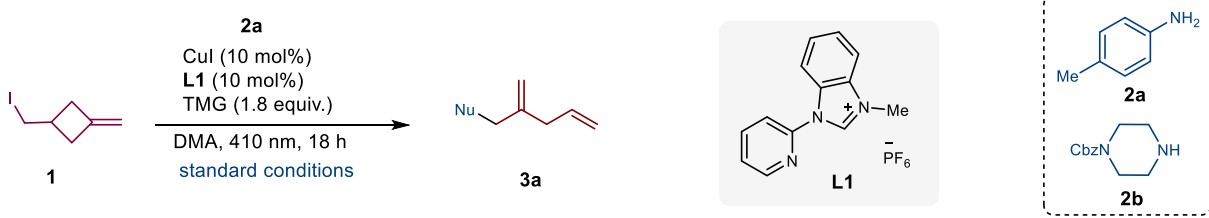

| Entry | Variant of standard conditions | NMR yield | Entry | Variant of standard conditions | NMR yield |
|---|---|---|---|---|---|
| 1 | None | 35% | 6 | CuI (20 mol%) + KBr (1 eq) | 49% |
| 2 | CuI (20 mol%) | 45% | 7 | CuI (50 mol%) | 86%(85%)[a] |
| 3 | CuI (20 mol%) + NaI (1 eq) | 72% | 8 | CuI (100 mol%) | 83% |
| 4 | CuI (20 mol%) + KI (1 eq) | 69% | 9 | without CuI, L1 or TMG | N.D. |
| 5 | CuI (20 mol%) + TBAI (1 eq) | 68% | 10 | In dark | N.D. |

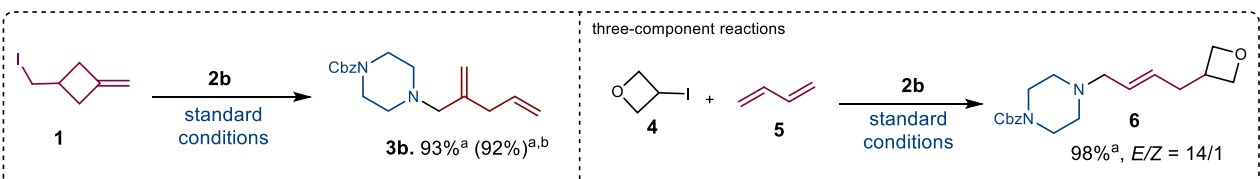

**Fig. 1 | Strategies to access allylic amines and copper(III) complex for bond formation. a** C-N bond formation via η[3] allyl metal complex. **b** Facilitated inner-sphere reductive elimination of copper(III) complex. **c** Strategy to achieve C-N bond formation via allyl copper complex and rational design of radical pathways. **d** Optimal condition to access allylic amines via two or three-component reactions at 0.1 mmol scale. TMG Tetramethylguanidine, TBAI tetrabutylammoniumiodide. [a]isolated yield, [b]5 mmol.

allyl-Cu(III)-Nu complexes when using "hard" nucleophiles[29–35]. However, this paradigm presents inherent limitations when using soft nucleophiles such as alkyl or aryl amines, as it requires the use of a very strong base for the deprotonation of NuH.

We therefore hypothesized a Cu(I) complex, devoid of pre-coordination to the deprotonated-nucleophile (Nu⁻), could facilitate the radical generation of allylic Cu(III) intermediates through oxidative addition to C(sp3)-X bonds, thereby enabling efficient Tsuji-

Trost-type allylic aminations. Our strategy leverages the unique photoredox properties of pyridyl-carbene (Py-NHC) Cu(I) complexes, which facilitate alkyl halide activation via halogen atom transfer (XAT) to generate alkyl radicals and Cu(II) intermediates[36]. Through rational design of radical pathways, we envisioned two distinct routes to allylic Cu(III) species: (1) radical induced cyclobutane ring-opening rearrangements[37–39], and (2) radical addition of transient alkyl radicals to 1,3-dienes[40–42] (Fig. 1c). Subsequent radical rebound to Cu(II)

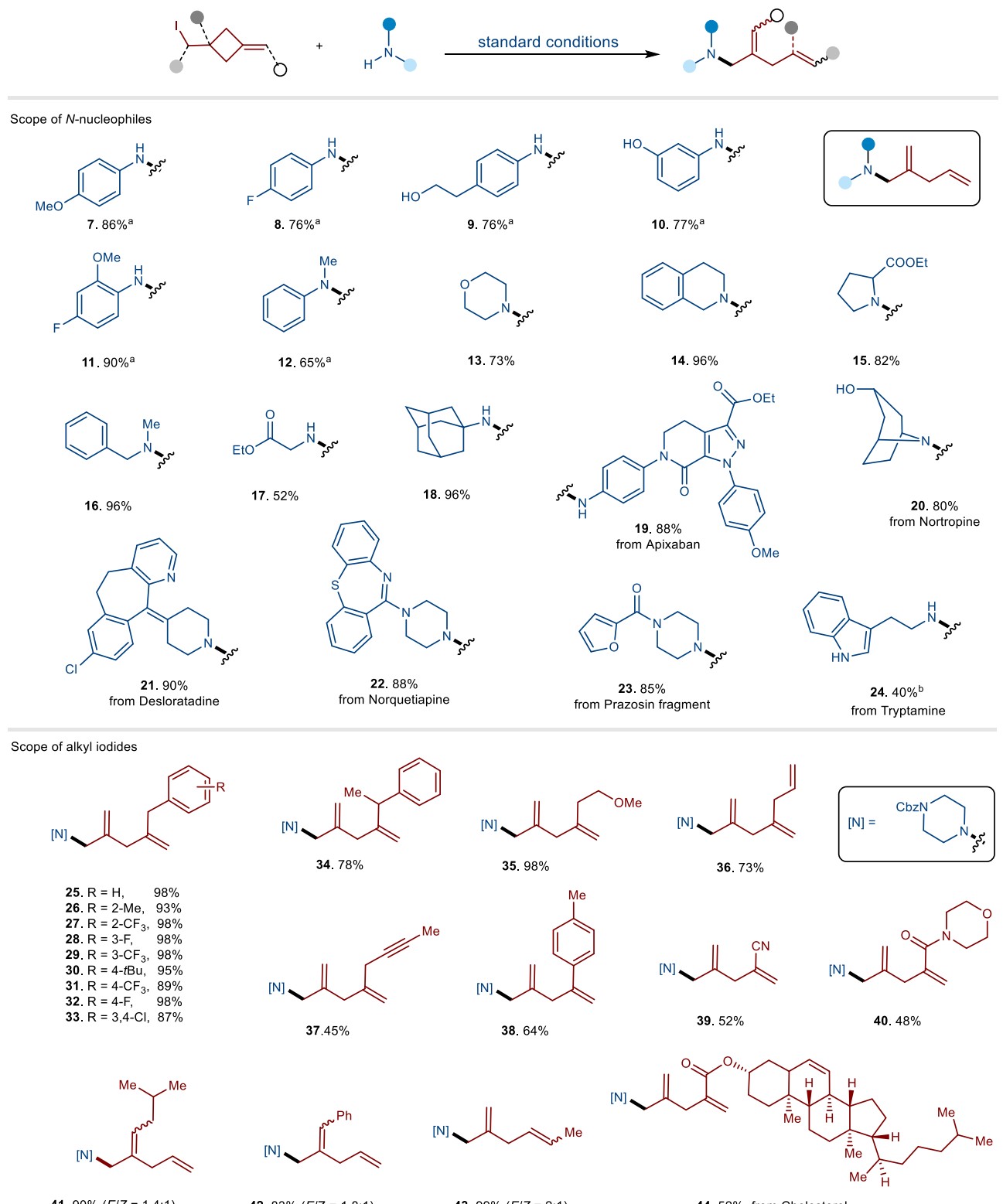

**Fig. 2 | The substrate scope of *N*-nucleophiles and alkyl iodides.** The reactions were run using *N*-nucleophiles (0.1 mmol, 1.0 equiv) and alkyl iodides (0.15 mmol, 1.5 equiv). [a]50 mol% CuI was used. [b]*N*-nucleophiles (0.2 mmol) and alkyl iodides (0.1 mmol) were used.

centers would generate allylic Cu(III) complexes primed for nucleophilic substitution by amines, thereby establishing a unified platform for divergent allylic amine synthesis[43].

In this study, we disclose a copper-photocatalyzed paradigm enabling two complementary approaches to allylic amines involving outer-sphere nucleophilic substitution mechanism at allylic copper(III)

intermediates. The first approach provides access to synthetically challenging skipped dienyl amines[44] through two-component reaction from cyclobutyl methyl iodides and amines. The second method achieves stereoselective 1,4-carboamination of 1,3-dienes using alkyl iodides as carbon sources, delivering functionalized allylic amines with high *E*/*Z* selectivity (up to 19:1). Notably, our system demonstrates

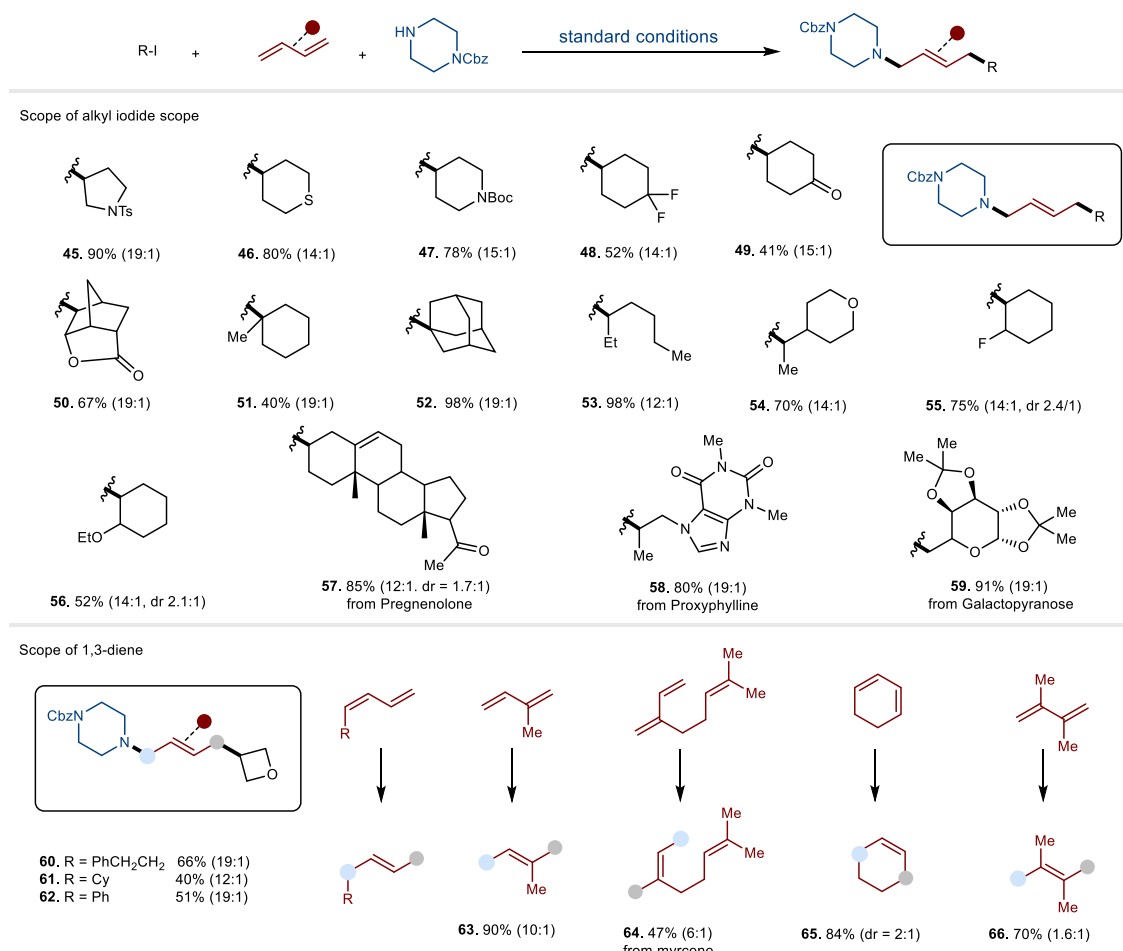

**Fig. 3 | The substrate scope of alkyl iodides and 1,3-dienes.** The reactions were run using *N*-nucleophiles (0.1 mmol, 1.0 equiv), 1,3-dienes (0.3 mmol, 3.0 equiv) and alkyl iodides (0.15 mmol, 1.5 equiv). The ratio of the two isomers (*E* and *Z*) is provided inside parentheses right after the compound's yield unless otherwise noted. dr diastereoisomer ratio.

good functional group tolerance across diverse amine substrates (alkyl, aryl) and electrophilic partners, underscoring its broad synthetic utility. Mechanistic investigations combining kinetic experiments and DFT calculations reveal the involvement of an outer-sphere nucleophilic substitution to form C-N bond at allylic copper(III) complex.

## Results

### Optimization of photoinduced copper-catalyzed allylic amines synthesis

At the outset of this investigation, to streamline the optimization process, we selected a two-component reaction as the model system for synthesizing allylic amines featuring skipped double bonds. This transformation is proposed to proceed via a radical-induced ring-opening mechanism, generating an allylic radical intermediate that subsequently forms the allylic copper(III) complex. The reaction employed an in situ-generated Py-NHC-Cu(I) complex[36] as the catalyst, 1-(iodomethyl)−3-methylenecyclobutane (**1**) as the allylic radical precursor, and *para*-methyl aniline (**2a**) as the nucleophile. A comprehensive screening of reaction parameters, including the copper source, ligand, solvent, base, and additives, was conducted to identify the optimal conditions (Supplementary Tables 1–6). Notably, while the choice of base and solvent significantly influenced the reactivity (Supplementary Tables 2 and 3), control experiments (Supplementary Table 6 & Fig. 1a) revealed that the ligand, light, and copper source were indispensable for maintaining catalytic activity. It is important to emphasize that excess iodide anion plays a critical role in maintaining

high reactivity for the formation of skipped dienyl amine **3a** (Fig. 1d, Entries 1-8). The additional iodide likely suppresses ligand exchange with the aniline nucleophile **2a** in the presence of TMG base, thereby preventing the direct coupling between substrate **2a** and alkyl radical **1a** without ring-opening (Supplementary Table 1). After systematic optimization, the following conditions were identified as optimal: Py-NHC ligand (**L1**), CuI as the copper source, 1,1,3,3-tetramethylguanidine (TMG) as the base, and dimethylacetamide (DMA) as the solvent. Under these conditions, the target allylic amines **3a**, **3b**, and **6** were obtained in excellent isolated yields of 85%, 93%, and 98%, respectively, with high stereoselectivity (*E*/*Z* = 14/1) for **6** (Fig. 1d). we also run the reaction at 5 mmol scale to deliver final product **3b** in 92% yield without compromising the reactivity.

### Substrate scope

Various *N*-nucleophiles were tested to investigate the generality of the cascade C-C bond cleavage/C-N coupling reactions (Fig. 2). Aniline-type nucleophiles with different functional groups demonstrated compatibility, yielding the desired products (**7-11**) in good to excellent yields. Notably, compounds bearing free hydroxyl groups (**9, 10**) did not inhibit reactivity and were transformed efficiently into the corresponding products without any detectable formation of undesired C-O coupling side products. Under optimal conditions, *N*-alkyl anilines were efficiently converted to the corresponding 1,4-skipped dienyl amines (**12**). Both primary and secondary alkyl amines, including cyclic and acyclic varieties, exhibited excellent reactivity (**13-24**). Functional groups such as ester (**15, 17**), amide (**19**) and chlorine (**21**) were well

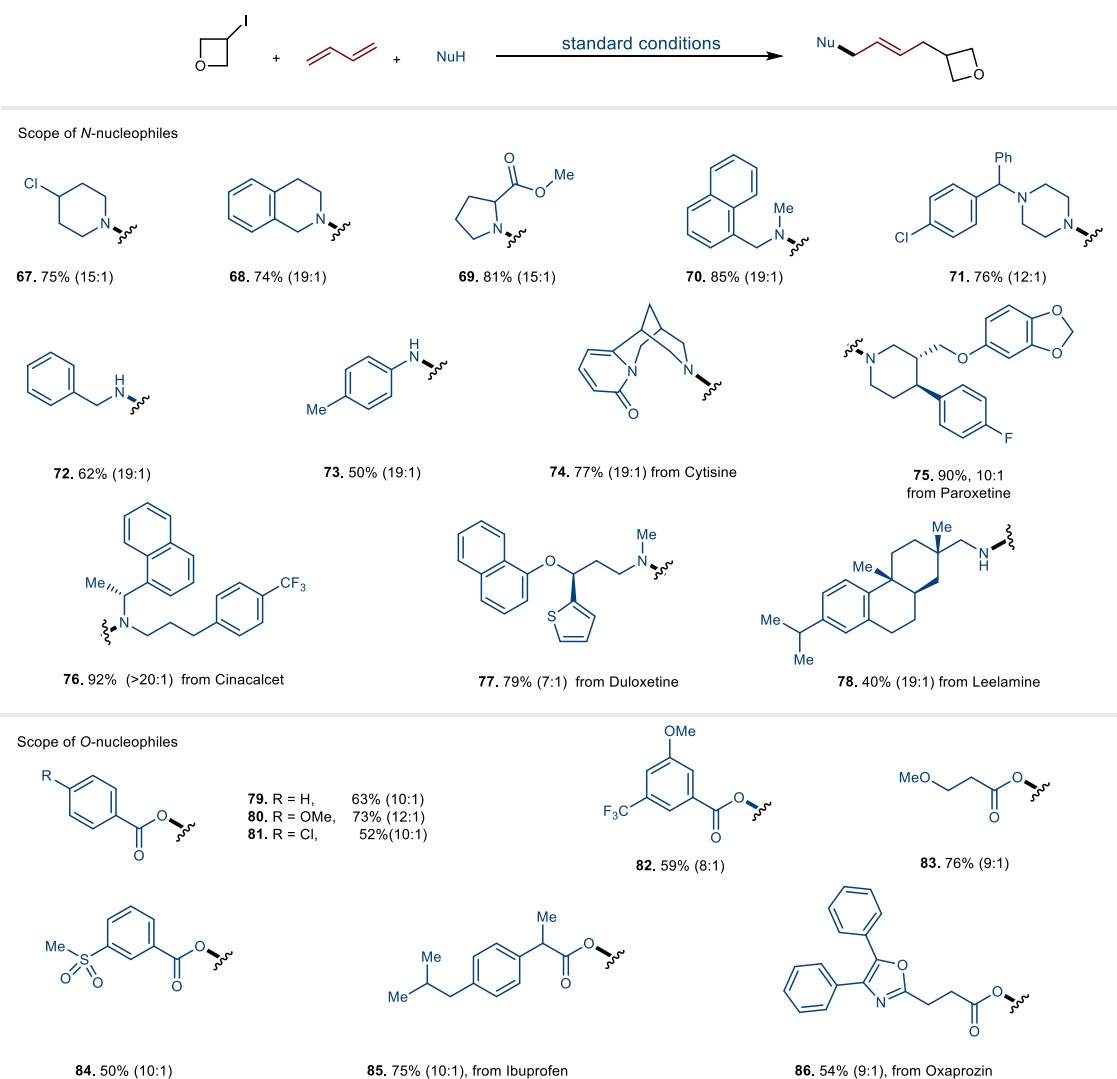

**Fig. 4 | The substrate scope of nucleophiles.** The reactions were run using *N/O*-nucleophiles (0.1 mmol, 1.0 equiv), 1,3-dienes (0.3 mmol, 3.0 equiv) and alkyl iodides (0.15 mmol, 1.5 equiv). The ratio of the two isomers (*E* and *Z*) is provided inside parentheses right after the compound's yield unless otherwise noted.

tolerated in these reactions. Furthermore, some pharmaceutical molecules and their fragments were also subjected to optimal conditions, providing the corresponding products in high yield (**19-24**). Various heterocycles, including pyridinone (**19**), pyridine (**21**), dibenzothiazepine (**22**), furan (**23**), and free indole (**24**), proved to be compatible with this approach.

We subsequently evaluated the scope of the reactions using a diverse array of cyclobutyl alkyl iodides with various substitutions (Fig. 2). Substrates featuring functionalized substitutions on the cyclobutyl ring displayed high reactivity, yielding products (**25-44**) with good to excellent yields. Functional groups such as trifluoromethyl (**27, 29**), fluoride (**28, 32**), chloride (**33**), and cyanide (**39**) were compatible with the cascade reactions. Alkyl iodides containing unsaturated double (**36**) and triple bonds (**37**) effectively generated the corresponding polyene and polyenyne, respectively, within our catalytic system. Notably, products **39** and **40**, which contain conjugated double bonds, were compatible under optimal conditions and could be isolated as pure compounds without the formation of byproducts through radical addition to the conjugated double bond. Substrates with substitutions on the double bonds can result in the formation of *E*/*Z* isomer products due to radical induced non-selective C−C bond cleavage (**41, 42**). Secondary iodide proceeded well, yielding product (**43**) with moderate regioselectivity. Steroid derivative iodide (**44**) can perform very well in standard conditions.

We moved to investigate the scope of three-component reactions to access allylic amines (Fig. 3). The catalytic system demonstrated broad substrate compatibility, effectively accommodating primary, secondary, and tertiary alkyl iodides, all exhibiting good to excellent reactivity with high *E*/*Z* ratios. A variety of functional groups−including esters, ethers, carbonyls, thiol ethers, Boc (*t*-butoxy carbonyl), heterocycles, free alcohols, and tosylates−were well tolerated (**45-56**). Furthermore, complex alkyl iodides were successfully employed as radical sources, ultimately leading to the production of 1,4-carboamination compounds (**57-59**) under consistent catalytic conditions. Next, we investigated both cyclic and acyclic dienes with varying substitutions (**60-66**). Mono-substituted acyclic dienes exhibited good compatibility, yielding desired products with favorable regioselectivity and stereoselectivity (**60-62**). Dienes with additional substitutions also effectively produced final products, albeit resulting in worse stereoselectivity (**64-66**).

Both primary and secondary alkyl amines, as well as aromatic amines, proved to be suitable nucleophiles for generating final 1,4-carboamination products (**67-73**) with good yields (Fig. 4). The catalytic process was also applicable to pharmaceutical derivative amines, yielding 1,4-carbonaminated products (**74-78**) in high yield. These results clearly indicate that three-component reactions are quite general for accessing allylic amines through the nucleophilic substitution of the allyl copper complex within the catalytic cycle. We also explored

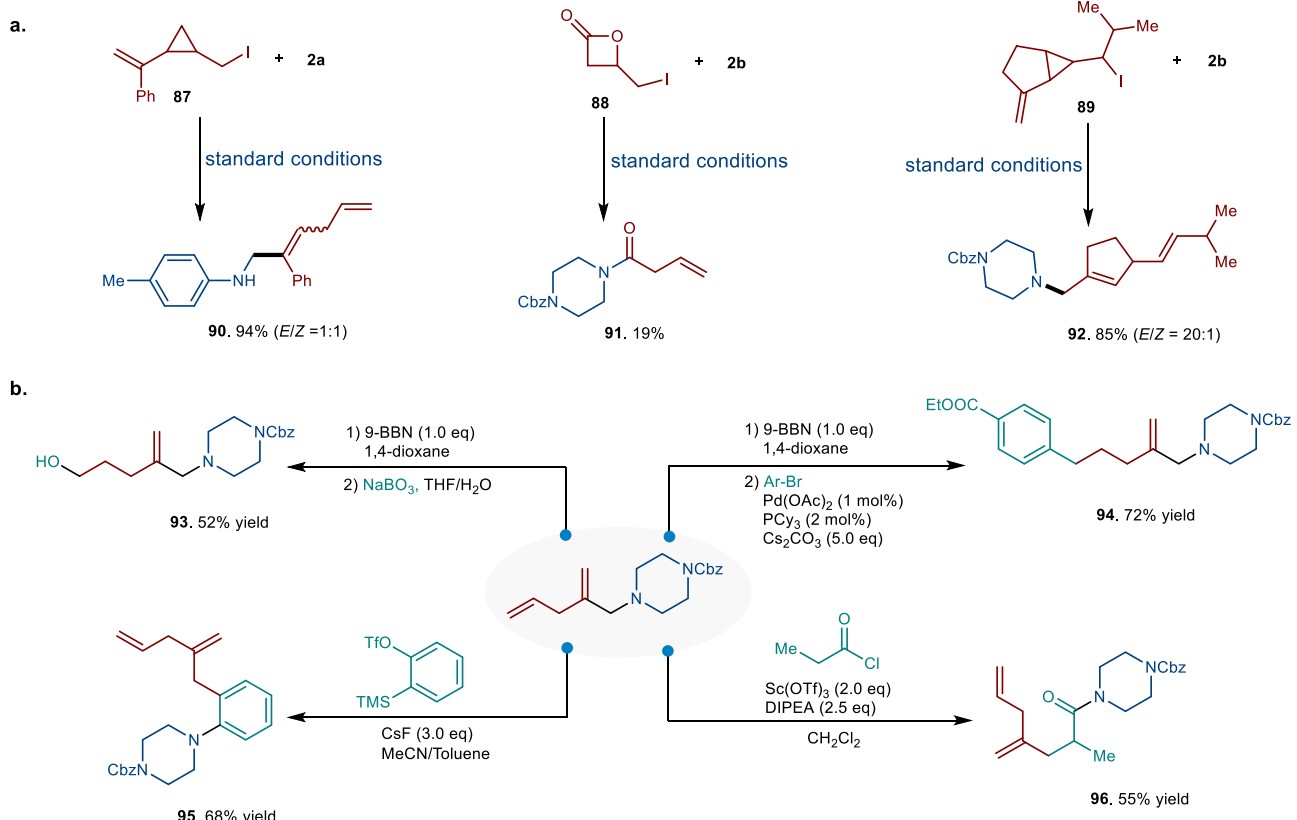

**Fig. 5 | Extension to alkyl iodides with various ring sizes and transformation of the skipped diene. a** Reactions of cyclic alkyl iodides (with various ring sizes) with amines under standard conditions. **b** The chemical transformation of skipped dienes. 9-BBN 9-Borabicyclo[3.3.1]nonane. DIPEA *N,N*-Diisopropylethylamine.

the use of carboxylic acids as nucleophiles under the standard conditions (**79-86**). Notably, a variety of both alkyl and aromatic carboxylic acids proved to be effective participants in the three-component reactions.

## Further application and transformation
Under the standard reaction conditions, various substrates with different ring sizes were evaluated (Fig. 5a). Notably, the highly strained three-membered ring substrate **87** reacted efficiently to afford the desired product **90** in good yield. Interestingly, these same conditions unexpectedly converted lactone derivative **88** to allylic amide **91**, albeit in low yield. Particularly impressive was the transformation of substrates containing a bicyclo[3.1.0] scaffold, which provided the corresponding product **92** in excellent yield (85%). However, substrates featuring less strained five- or six-membered rings failed to undergo the desired transformation under these conditions.

Furthermore, 1,4-skipped dienyl amines are not only significant components of alkaloids but can also be transformed into other valuable compounds. These amines are amenable to various derivatization reactions (Fig. 5b). For instance, selective borylation of the terminal double bond facilitates the formation of a versatile boronate ester building block, which can be easily converted into a free alcohol (**93**) or undergo Suzuki coupling to introduce an aryl group (**94**). The acyl-Claisen and benzyne Aza-Claisen reactions of allylic amines represent powerful methodologies for reconstructing functionalized amines. The resulting dienyl amines exhibited high reactivity in forming new diene compounds (**95**, **96**) through both acyl-Claisen and benzyne Aza-Claisen reactions.

## Mechanism studies
To gain insight into the copper catalytic cycle, we focused on a two-component reaction to explore the underlying mechanisms. This section examines key catalytic steps, including the electron transfer process, the formation of radical fragments, the generation of radical species from the allylic copper complex, and nucleophilic substitution, utilizing both experimental methods and DFT calculations.

**Investigation of single electron transfer process between photo-excited copper complex with alkyl iodide.** Ultraviolet-visible absorption experiments confirmed that the in-situ generated Py-NHC Cu(I) complex, formed from CuI, PyNHC, and TMG, serves as the active light-harvesting species, with additional reactants not affecting the absorption spectra (Supplementary Fig. 3). Subsequently, we measured transient absorption (TA) spectra over nanosecond to sub-microsecond timescales to investigate the electron transfer process between copper catalyst and alkyl iodide **1**. Notably, we observed that the long-lived (2 μs) [3]MLCT state of the Cu(I) complex can be quenched by the corresponding alkyl iodide (**1**) (Fig. 6a, b). The plot of the triplet excited state lifetime against the alkyl iodide concentration demonstrates non-linear Stern-Volmer quenching behavior (Fig. 6c), which aligns with the inner-sphere electron transfer process involving an exciplex formed between the [3]MLCT state Cu(I) complex and the alkyl iodide[36].

**Radical formation and capture reactions.** The experiment using 2,2,6,6-tetramethylpiperidinyl-l-oxide (TEMPO) as the radical trapper confirmed the formation of the cyclobutyl methyl radical from alkyl iodide **1**, which reacted with TEMPO to form adduct (**97**). The TEMPO adduct of the 1,4-skipped dienyl radical was not observed, resulting in the complete inhibition of the reaction. This suggests that the radical-induced cyclobutyl ring-opening process can barely compete with the direct radical coupling with TEMPO. (Fig. 6d). To further investigate the radical-induced ring-opening process, the neutral radical trapper diphenyl disulfide was introduced as an additive to the reaction mixture, resulting in the formation of three products: the direct coupling

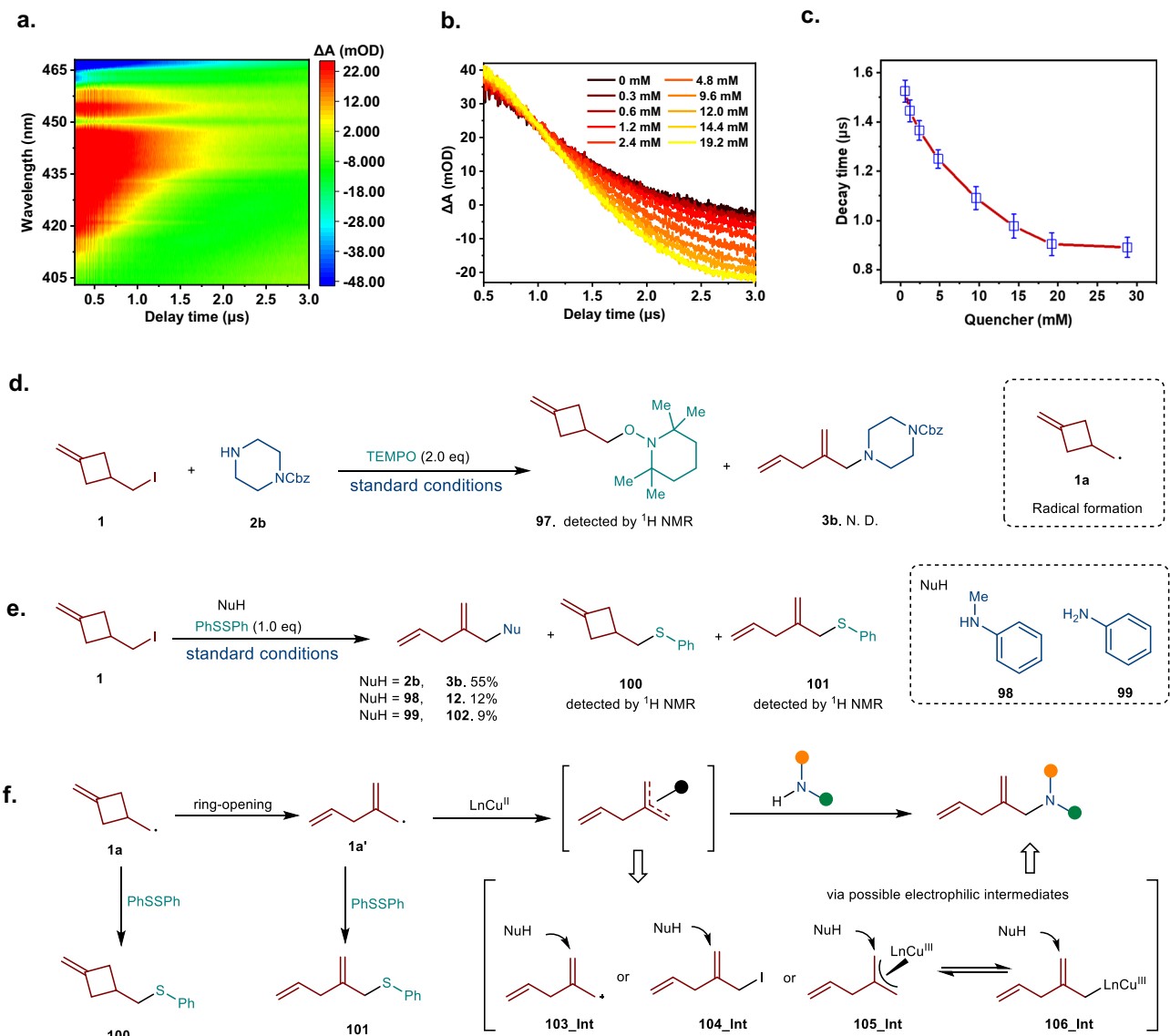

**Fig. 6 | Mechanistic studies. a** Two-dimensional pseudo-color nanosecond TA spectra of the in-situ generated copper complex. (λpump = 355 nm) **b** TA kinetics at 440 nm of the in-situ generated copper complex. **c** TA lifetime plotted as a function of the quencher concentration. Bars represent mean ± SD (*n* = 20; individual data points are plotted). **d** Radical trapping experiment. **e** Studies of radical induced ring-opening process. **f** Plausible intermediates involved in reaction.

product (**100**), the ring-opening C-S coupling product (**101**), and the desired 1,4-skipped dienyl amines (**3b, 12**, or **102**) (Fig. 6e). These results demonstrate that the cyclobutyl methyl radical can undergo ring-opening to generate the 1,4-skipped dienyl radical, which is available for subsequent C-N couplings. Furthermore, we noted a decrease in the yield of skipped dienyl amines, as the nucleophilicity of the substrates diminished. Thus, the nucleophilic substitution step was involved in C-N bond formation process via possible four electrophilic intermediates, i.e., allylic cation **103_Int**, alllylic iodide **104_Int**, π-allylcopper complex **105_Int**, and σ-allylcopper complex **106_Int** (Fig. 5f).

**Exploration of the nucleophilic substitution step for C-N couplings.** To gain insight into the nucleophilic substitution process, we conducted kinetic studies on the reaction of PhNHMe (**88**) and alkyl iodide (**1**) (Fig. 7a). The kinetics revealed a zeroth-order dependence on alkyl iodide (**1**), while the reaction exhibited first-order dependence on both the copper catalyst and the amine (**98**) (Fig. 7b, c). This suggests that both the copper catalyst and the amine are involved in the turnover-limiting step, specifically the nucleophilic substitution step.

Subsequently, we performed kinetic experiments to evaluate the electronic effects of *para*-substitution in *N*-methyl anilines (**98X**) as the nitrogen source (Fig. 7d). The negative slope (ρ = −0.853) of the Hammett plot indicates that stronger nucleophilic amines increase the rate of *N*-nucleophilic attack in C−N coupling reactions. The small negative ρ value also suggests that the substitution step follows an $S_N2$ mechanism rather than an $S_N1$ mechanism; otherwise, a ρ value less than −5 would be expected[45]. Additionally, the calculated oxidation potential of the skipped dienyl radical is 0.94 V versus Ag/AgCl, while the reduction potential of the Cu(II) complex is 0.48 V versus Ag/AgCl, consistent with the cyclic voltammogram measurements (0.41 V vs Ag/AgCl) (Fig. 7e). Thus, the skipped dienyl radical is unlikely to undergo single electron oxidation by the Cu(II) complex. Collectively, these results support the conclusion that the nucleophilic substitution occurs via an $S_N2$ mechanism.

**Density functional theory calculations.** We aimed to elucidate the nucleophilic addition step involved in the formation of C−N bonds using density functional theory (DFT) calculations (Fig. 8). Based on experimental data and literature[36], it was demonstrated that Py-NHC-

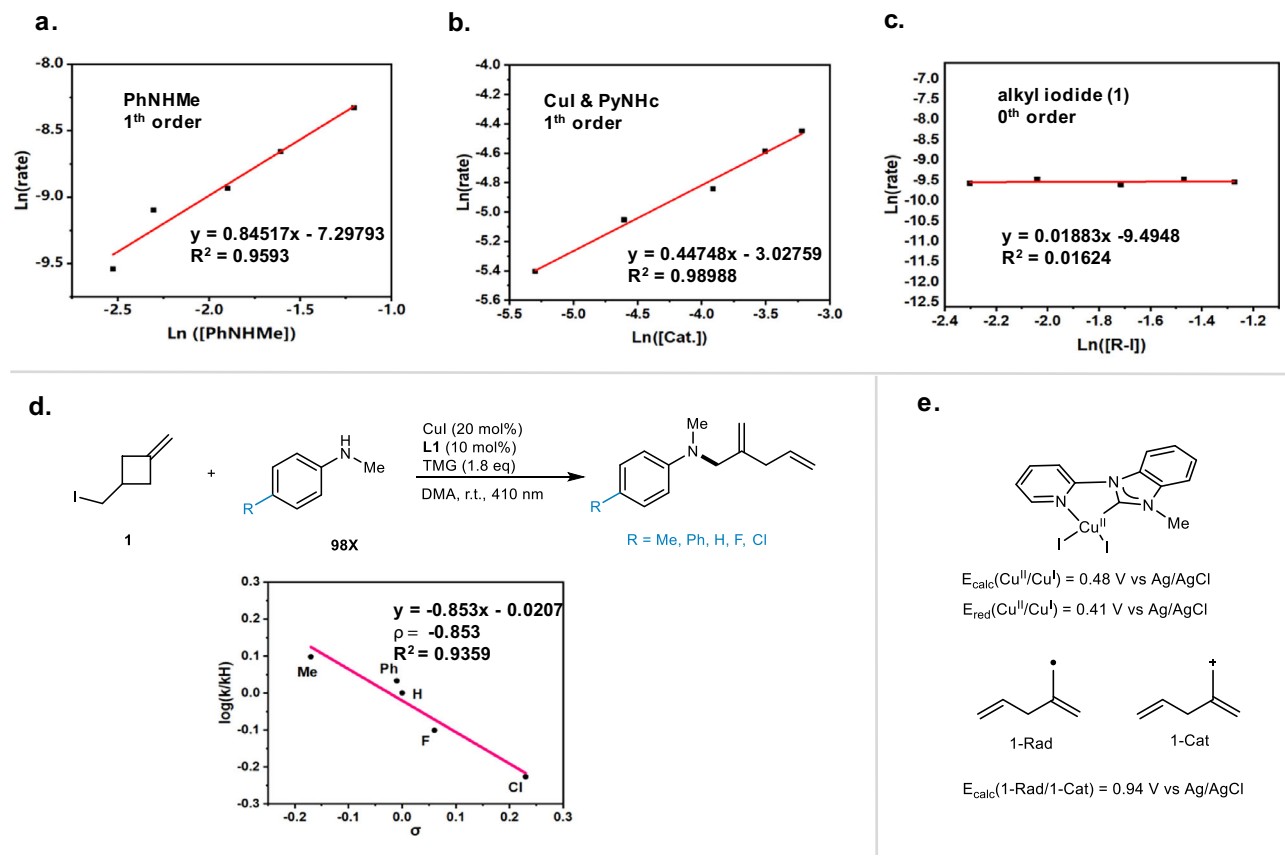

**Fig. 7 | Investigation of C-N bond formation. a** Reaction Kinetic studies of PhNHMe. **b** Reaction Kinetic studies of CuI and PyNHC. **c** Reaction Kinetic studies of alkyl iodide (**1**) **d** Hammett plots of reaction components. **e** Redox potential of intermediates.

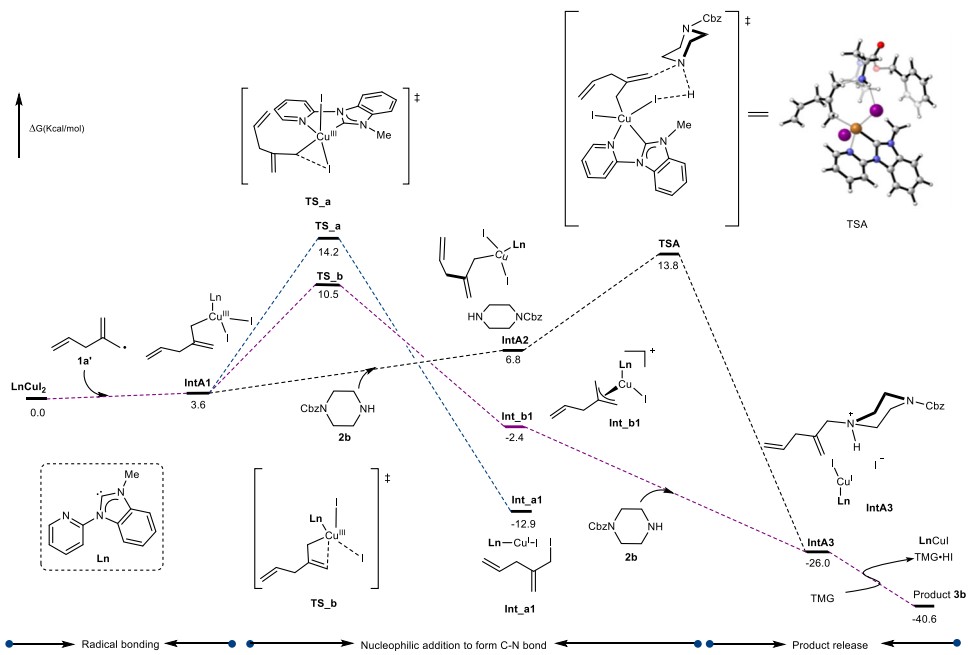

**Fig. 8 | Gibbs free energy profile at the level of B3LYP-D3/Def2TZVP at 298.15 K.** The solvent effects of *N,N*-dimethylacetamide by the PCM model was included. The energies are given in kcal/mol.

Cu(I) abstracts the iodide atom from compound **1** under irradiation with a 410 nm LED, generating a Cu(II) complex and the allylic radical **1a'** (Fig. 6). Therefore, our calculations were initiated from the Py-NHC-Cu(II) complex and the allylic radical **1a'**. The Cu(II) complex captures

the skipped dienyl radical, resulting in the formation of an 18-electron saturated σ-allylcopper complex (**IntA1**). The σ-allylcopper complex (**IntA1**) readily undergoes an outer-sphere $S_N2$ reaction via a 7-membered ring transition state (**TSA**), overcoming an energy barrier

**Fig. 9 | Proposed plausible mechanism for photocopper catalysis.** TMG Tetramethylguanidine. XAT Halogen-Atom Transfer.

of 10.2 kcal/mol to form the final product. However, **IntA1** preferentially isomerizes to the thermodynamically more stable π-allylcopper complex (**Int_b1**), which then participates in a barrier-free outer-sphere S$_N$2 reaction to afford the product (Supplementary Fig. 14). These computational findings are consistent with kinetic experimental data using **2b** as the nucleophile, which exhibited zeroth-order dependence on nucleophile concentration (Supplementary Fig. 7). Notably, the generation of allylic iodide (**IntA1→Int_a1**, 10.6 kcal/mol), which could potentially consume the N-nucleophile, is still less favorable than the catalytic allylic substitution pathway (**IntA1 → TS_b→Int_b1→IntA3**). Additionally, inner-sphere pathways involving Cu(III) intermediates for C-N bond formation require traversing a higher-energy transition state (**TSA**, ΔG$^‡$ = 12.3 kcal/mol, Supplementary Fig. 13), supporting the favorability of the outer-sphere S$_N$2 catalytic route.

**Proposed mechanism.** Based on comprehensive experimental investigations, DFT calculations, and supporting literature[36], we propose a plausible integrated reaction mechanism as depicted in Fig. 8. The catalytic cycle initiates with the in situ formation of a Py-NHC copper(I) complex (**A**), which undergoes photoexcitation under blue LED irradiation to form the excited-state complex **A***. This activated species coordinates with the alkyl iodide substrate and abstracts the iodide moiety via an inner-sphere single electron transfer process[36,46,47] (Fig. 6a–c). The resulting alkyl radical intermediate (**1a**) subsequently undergoes radical fragmentation to generate an allylic radical species (**1a'**) (Fig. 6d–f). The allylic radical **1a'** then coordinates with the copper(II) complex **B** to form allylic copper intermediates **C** or **C'** (Fig. 7 & 8). DFT calculations coupled with kinetic studies suggest that intermediate **C'** preferentially undergoes outer-sphere nucleophilic substitution to facilitate C-N bond formation[43], ultimately releasing the final product while regenerating the

copper complex **A** in Fig. 9. Nevertheless, alternative mechanistic pathways, including inner-sphere nucleophilic substitution, cannot be ruled out, particularly when employing different nucleophilic partners in three-component reactions. For example, in three-component reaction systems employing TMG (1,1,3,3-tetramethylguanidine) as a base to deprotonate nucleophiles (e.g., carboxylic acids)[48–50], the reaction mechanism might shifts toward an inner-sphere nucleophilic substitution pathway (Supplementary Fig. 15). This alternative mechanism operates stereoselectively, preferentially yielding products with E-configuration (Supplementary Fig. 16 and 17). Additionally, for the outer-sphere S$_N$2 reaction proceeding via the allylic copper(III) intermediate pathway with amines as nucleophiles, the thermodynamically favored π-allylcopper complex acts as the key intermediate Cu$^{III}$-yita3E in Supplementary Fig. 18, dictating the selective formation of E isomers via a barrier-less route.

## Discussion
In this study, we developed a photocopper catalysis system for the synthesis of allylic amines through C−N coupling via outer-sphere nucleophilic substitution at allylic copper(III) complexes. The photo-excited copper(I) complex effectively abstracts the iodide atom from alkyl iodides, generating the corresponding alkyl radicals. These radicals subsequently convert into allylic radicals via radical rearrangement and addition to conjugated 1,3-dienes, ultimately recombining with the Cu(II) complex to form allylic copper(III) intermediates. The C−N bond formation through the outer-sphere S$_N$2 pathway may inspire the development of new copper catalysis methods. The broad substrate scope exhibited in our experiments highlights the practical applicability of this copper catalytic system, particularly its effectiveness in coupling pharmaceutical derivatives. Given the versatility of

our system, we anticipate considerable interest from the medicinal chemistry, synthetic chemistry, and catalysis communities.

## Methods

### General procedure for the two-component reactions to access allylic amines

To an oven-dried 10 mL reaction vial were added CuI (1.9 mg, 0.01 mmol, 10 mol% or 9.5 mg, 0.05 mmol, 50 mol%), PyNHC (3.6 mg, 0.01 mmol, 10 mol%), 1,1,3,3-Tetramethylguanidine (TMG) (23 μL, 0.18 mmol, 1.8 equiv), and DMA (1 mL) in a nitrogen-filled glove box. The resulting mixture was stirred for 10 min, followed by adding N-nucleophiles (0.1 mmol, 1.0 equiv) and alkyl iodides (0.15 mmol, 1.5 equiv) in sequence, and sealed with a screwed cap. The sealed vial was placed on a photo-reactor under irradiation of LEDs (410 nm, 6 W). The mixture was stirred at room temperature for 18 h, quenched with $H_2O$, and extracted with ethyl acetate. The combined organic layers were dried over anhydrous $Na_2SO_4$, and concentrated in vacuo. The crude product was purified by silica gel column chromatography to afford the coupling product.

### General procedure for three-component reactions to access allylic amines

To an oven-dried 10 mL reaction vial were added CuI (1.9 mg, 0.01 mmol, 10 mol%), PyNHC (3.6 mg, 0.01 mmol, 10 mol%), 1,1,3,3-tetramethylguanidine (TMG) (23 μL, 0.18 mmol, 1.8 equiv) and DMA (1 mL) in a nitrogen-filled glove box. The resulting mixture was stirred for 10 min, followed by adding N-nucleophiles (0.1 mmol, 1.0 equiv), 1,3-dienes (0.3 mmol, 3.0 equiv) and alkyl iodides (0.15 mmol, 1.5 equiv) in sequence, and sealed with a screwed cap. The sealed vial was placed on a photo-reactor under irradiation of LEDs (410 nm, 6 W). The mixture was stirred at room temperature for 18 h, quenched with $H_2O$, and extracted with ethyl acetate. The combined organic layers were dried over anhydrous $Na_2SO_4$, and concentrated in vacuo. The crude product was purified by silica gel column chromatography to afford the coupling product.

## Data availability

All data that support the findings of this study are available within the paper, its supplementary information files and Supplementary Dataset 1-3 to this manuscript. All data are available from the corresponding author upon request.

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

## Acknowledgements

The authors acknowledge financial support from the National Natural Science Foundation of China (No. 22371027) and State Key Laboratory of Elemento-Organic Chemistry, Nankai University (202409). K.W. acknowledges financial support from the Chinese Academy of Sciences (YSBR-007) and the New Cornerstone Science Foundation through the XPLORER PRIZE. M.G. acknowledges financial support from JSPS KAKENHI (Grant Numbers JP24H02219) in Transformative Research Areas (A) JP24A202 Integrated Science of Synthesis by Chemical Structure Reprogramming (SReP). The computation results in this work are the outcomes achieved through the MANABIYA (ACADEMIC) program conducted by Institute for Chemical Reaction Design and Discovery (ICReDD), Hokkaido University, which was established by World Premier International Research Initiative (WPI), MEXT, Japan. A part of the calculation results was computed at the Research Center for Computational Science, Okazaki, Japan (Project Number 22-IMS-C002) and the computer center of Kyoto University. The authors also acknowledge the assistance of Dr. Huihui Wan in DUT Instrumental Analysis Center for HRMS analysis.

## Author contributions

L.L. conceived the research project. L.L. and H.L. designed the catalytic reactions. H.L., Y.M. and F.Y. contributed to optimizing the coupling reactions and mechanistic studies. H.L., Y.M., F.Y. and S.X. performed experiments of catalytic reactions. Y.Y. and H.L. performed experiments of TA measurements and analysis. M.G., L.L., H.J. and Y.L. performed the calculation. L.L., K.W. and M.G. co-directed the research project. L.L., K.W., M.G., H.L., S.X. and Y.Y. prepared the manuscript. All authors contributed to discussions on the manuscript.

## Competing interests

The authors declare no competing interests.
