## [Transparent Peer Review file · Nature Communications]

Light-Driven Radical Copper-Catalyzed Allylic Amination via Allylic Copper Intermediates

Corresponding Author: Professor Luqing Lin

Version 0:

Reviewer comments:

Reviewer #1

(Remarks to the Author)

This is a difficult case! The manuscript titled "Light-Driven Radical Copper-Catalyzed Allylic Amination via Allylic Copper Intermediates" presents a copper-photocatalyzed method for allylic amination through allylic copper(III) intermediates. The copper complex has been discovered by their own group (Nature Communications 2024, 15, 5647, Org. Chem. Front., 2024, 11, 6380-6384 and OL, 10.1021/acs.orglett.5c00602), which could reduce the alkyl iodide to generate alkyl radical intermediates and applied in the C-N coupling, heck reaction and radical addition. The only different in this paper is that they successfully generated allylic radical from allylic iodides or alkyl iodide with 1,3-diene. However, similar synthetic approaches to form allylic copper intermediate through radical pathway has been reported by several groups, such as (ACIE, 2021, 10.1002/anie.202110084). The only different is that 1,4-carboamination. However, radical 1,4-carboamination has also been achieved by palladium or cobalt catalysis. The authors indeed explored the good substrate scope and mechanistic studies and DFT calculation. Thus, this is a difficult case, I prefer to reject it in the current version. the following comments may be considered for further development:

1. To broaden applicability, the authors should explore other ring sizes and heteroatom-containing substrates.
2. I am satisfied with the broad substrate scope except the discovery of drug-like molecules as nucleophiles. The scope should be expanded to better reflect the prevalence of nitrogen-containing drugs and natural products. For instance, nucleophiles derived from β -lactams, alkaloids, or kinase inhibitors would better demonstrate practical relevance. Authors should at least give different cases in the two different reactions.
3. There are few questions about the scope section: 1) Why is 50 mol% CuI required for aniline derivatives in two-component reactions (e.g., 7-11), while three-component reactions (e.g., 73) proceed with 10 mol%? Rationale for this difference should be provided; 2) why a simple ester group could lead to a not nice yield on 17 but was less affective on 69; 3) there is a potential chemo selectivity on Tryptamine, which may be able to explain low yield of 24. Authors should confirm whether indole alkylation byproducts form and discuss mitigation strategies.
4. The high E/Z selectivity is highlighted but not mechanistically explained. Authors should provide extra computational analysis of the steps leading to E/Z selectivity and discuss how the Copper complex geometry and/or radical rebound step influences the stereochemistry.
5. Manipulation must be developed to enhance the utilization potentiality of this work.
6. Light on/off experiment and/or quantum yield experiment should be done to make sure this work is not a radical chain reaction.
7. The proposed mechanism relies on the formation of allylic Cu(III) intermediates and the DFT calculation looks fine, however, direct detection of Cu(III) intermediates using EPR spectroscopy seems to be possible like in Inorg. Chem. 2023, 62, 5387-5399. This is not strictly required but authors are encouraged to do so.

Reviewer #2

(Remarks to the Author)

The manuscript by Lin and coworkers describes a photocopper-catalyzed system for the synthesis of allylic amines via C-N coupling through outer-sphere nucleophilic substitution at allylic copper(III) complexes. This reaction employs soft nucleophiles, such as alkyl or aryl amines, addressing the issues associated with the traditionally used hard nucleophiles.

Through mechanistic studies, the authors experimentally confirmed the radical-mediated formation of allylic copper complexes and provided evidence for the involvement of outer-sphere nucleophilic substitution at allylic copper(III) complexes. This paradigm could be the complement of established methods. I suggest that this manuscript can be very interesting for readers of Nat. Commun. and can be accepted after major points below have been addressed.

1 . There are some formatting inconsistencies in the manuscript. For example, in line 101, the nitrogen atom in "N-nucleophiles" is not italicized, and in line 131, "E/Z" in "high E/Z ratios" is not italicized. Please carefully check the manuscript and the Supporting Information (SI) for similar formatting issues.

2 . The manuscript lacks a reaction condition screening table. It is recommended that the authors extract and summarize key screening data from the Supporting Information (SI) to provide readers with a clearer understanding of the reaction conditions.

3 . The authors propose a reasonable outer-sphere nucleophilic substitution mechanism at allylic copper(III) complexes. However, based on a survey of recent literature (J. Am. Chem. Soc. 2024, 146, 9444–9454; Angew. Chem. Int. Ed. 2024, 63, e202405560), an alternative pathway could also explain product formation, involving nucleophile coordination to Cu(III) through ligand exchange, followed by reductive elimination. It is recommended that the mechanistic discussion be expanded to address this possibility. Additionally, I believe that the nucleophiles in the system are also likely to pre-coordinate with the copper complex, given the strong basicity of TMG in the system.

4 . In the field of copper-catalyzed allylic functionalization, several important studies are worth citing, such as: Angew. Chem. Int. Ed. 2021, 60, 22956–22962; Angew. Chem. Int. Ed. 2024, 63, e202405560; Nature Commun. 2024, 15, 1483.

Reviewer #3

(Remarks to the Author)
see the attachment for details

Version 1:

Reviewer comments:

Reviewer #1

(Remarks to the Author)
The manuscript has been improved after revision, I recommend the publication of the current manuscript.

Reviewer #2

(Remarks to the Author)
The reviewer considers that the authors have adequately addressed most of the issues, and therefore the manuscript merits publication. This study reports a photocopper-catalyzed system for the synthesis of allylic amines via C–N coupling through outer-sphere nucleophilic substitution at allylic copper(III) complexes, effectively overcoming the limitations associated with conventional hard nucleophiles, and is expected to attract broad attention in the future. I am very pleased to unreservedly recommend the publication of this manuscript in Nature Communications.

Reviewer #3

(Remarks to the Author)
I am still not convinced by the novelty of this paper but I respect the comments of other reviewers and editors. Despite of this, the authors have addressed most of the comments reasonably except the one about solvation effect in geometry optimization. It is stated that "The optimizer implemented in develop version of global reaction route mapping (GRRM) program⁵ were used for all the optimizations of intermediates, transition states (TSs) and intersystem crossing points." but whether and what solvent model is used for geometry optimization is still unknown. The authors even deleted the description of PCM solvation model in the new SI, does it mean that no solvation model was used for the computational study? That would not be acceptable. Improve to the description of computational details to make the computational results reproducible.

Point-by-point response

Reviewer #1 (Remarks to the Author):

This is a difficult case! The manuscript titled “Light-Driven Radical Copper-Catalyzed Allylic Amination via Allylic Copper Intermediates” presents a copper-photocatalyzed method for allylic amination through allylic copper(III) intermediates. The copper complex has been discovered by their own group (Nature Communications 2024, 15, 5647, Org. Chem. Front., 2024,11, 6380-6384 and OL, 10.1021/acs.orglett.5c00602), which could reduce the alkyl iodide to generate alkyl radical intermediates and applied in the C-N coupling, heck reaction and radical addition. The only different in this paper is that they successfully generated allylic radical from allylic iodides or alkyl iodide with 1,3-diene. However, similar synthetic approaches to form allylic copper intermediate through radical pathway has been reported by several groups, such as (ACIE, 2021, 10.1002/anie.202110084). The only different is that 1,4-carboamination. However, radical 1,4-carboamination has also been achieved by palladium or cobalt catalysis. The authors indeed explored the good substrate scope and mechanistic studies and DFT calculation. Thus, this is a difficult case, I prefer to reject it in the current version. the following comments may be considered for further development:

[Our response]: Thank you for your thoughtful comments and for recognizing the breadth of substrate scope, mechanistic studies, and DFT calculations in our work. While allylic copper intermediates have been employed for C-O/C-C bond formations, their application in C-N coupling—particularly via an outer-sphere nucleophilic addition pathway—remains underexplored in copper catalysis. This distinction is critical, as such mechanisms, though well-documented for noble metals (e.g., Pd, Ir), are scarcely reported for earth-abundant copper. Our study addresses this gap by demonstrating copper's unique ability to mediate outer-sphere amination, complementing the prevalent inner-sphere pathways in copper chemistry. The elucidation of an outer-sphere nucleophilic addition pathway for allylic copper complexes represents a significant advancement in copper catalysis, expanding the synthetic potential of copper catalysis.

1. To broaden applicability, the authors should explore other ring sizes and heteroatom-containing substrates.

[Our response]: Thank you very much for your suggestions. We evaluated substrates with varying ring sizes and found that strained small rings, including three- and four-membered systems, are viable as allylic precursors. Notably, the four-membered lactone undergoes radical-induced fragmentation to afford unexpected allylic amide **91** (Figure 5).

2. I am satisfied with the broad substrate scope except the discovery of drug-like molecules as nucleophiles. The scope should be expanded to better reflect the prevalence of nitrogen-containing drugs and natural products. For instance, nucleophiles derived from β -lactams, alkaloids, or kinase inhibitors would better demonstrate practical relevance. Authors should at least give different cases in the two different reactions.

[Our response]: Thank you very much for your suggestions. We have revised the manuscript and use different drug-like molecules in different reactions. Please see the detail in the Figure 2 and 4.

3. There are few questions about the scope section: 1) Why is 50 mol% CuI required for aniline derivatives in two-component reactions (e.g., 7–11), while three-component reactions (e.g., 73) proceed with 10 mol%? Rationale for this difference should be provided; 2) why a simple ester group could lead to a not nice yield on 17 but was less effective on 69; 3) there is a potential chemo selectivity on Tryptamine, which may be able to explain low yield of 24. Authors should confirm whether indole alkylation byproducts form and discuss mitigation strategies.

[Our response]: Thank you for your questions.

1) In two-component reactions, excess CuI possibly provide excess iodide anion to inhibit the direct coupling of the alkyl radical and aniline because the excess iodide anion would compete with the aniline coordinating to the LnCu^{2+} . Otherwise some direct coupling would occur (see the Supplementary Table 1). The detail see the following pathways.

2) The primary amine is the nucleophile to yield compound 17, because small amount of over-alkylation product can be detected. To avoid this side reaction, we reduced the amount of alkyl iodides. While the secondary amine would not generate over-alkylation product. That is why less yield was obtained for generation of compound 17 compared to compound 69.

3) There is no chemo-selectivity problem for 24. Again, the nucleophile is primary amine, leading to the lower yield due to over-alkylation product. Therefore, it is necessary to reduce the alkyl iodide loading to inhibit the side product.

4. The high *E/Z* selectivity is highlighted but not mechanistically explained. Authors should provide extra computational analysis of the steps leading to *E/Z* selectivity and discuss how the Copper complex geometry and/or radical rebound step influences the stereochemistry.

[Our response]: Thank you for your suggestion. We have done the DFT calculation to explain the *E/Z* selectivity (Supplementary Figure 14-17). For the carboxylic acid as the nucleophiles, the transition state **TS_E1** exhibits lowest free energy, which leading to *E* isomer from the η^3 - π -allylic via inner-sphere pathways Supplementary Figure 16. In the outer-sphere S_N2 attack via a barrier-less pathway, when using the amines as nucleophiles, the key intermediate Cu^{III} - η^3 leading to *E* isomer is favored in Supplementary Figure 18.

We also added some comments in manuscript.

[added in the manuscript] Nevertheless, alternative mechanistic pathways, including inner-sphere nucleophilic substitution, cannot be ruled out, particularly when employing different nucleophilic partners in three-component reactions. For example, in three-component reaction systems employing TMG (1,1,3,3-tetramethylguanidine) as a base to deprotonate nucleophiles (e.g., carboxylic acids)⁴⁶⁻⁴⁸, the reaction mechanism might shift toward an inner-sphere nucleophilic substitution pathway (Supplementary Figure 15). This alternative mechanism operates stereoselectively, preferentially yielding products with *E*-configuration (Supplementary Figure 16 and 17). Additionally, for the outer-sphere S_N2 reaction proceeding via the allylic copper(III) intermediate pathway with amines as nucleophiles, the thermodynamically favored π -allylcopper complex acts as the key intermediate Cu^{III} - η^3 in Supplementary Figure 18, dictating the selective formation of *E* isomers via a barrier-less route.

5. Manipulation must be developed to enhance the utilization potentiality of this work.

[Our response]: Thank you very much for your suggestions. We have extended this methodology to additional allylic precursors as substrates, and the transformations of the resulting skipped dienes are presented in Figure 5.

6. Light on/off experiment and/or quantum yield experiment should be done to make sure this work is not a radical chain reaction.

[Our response]: Thank you very much for your suggestions. Our light on/off experiments clearly demonstrate that continuous light irradiation is indispensable for the reaction progression (Supplementary Figure 12). As the manuscript mentioned that photochemical activation of the copper(I) complex is required for iodide abstraction. The resulting copper intermediate participates in the key C-N bond formation step. These findings collectively rule out a radical chain propagation mechanism for this transformation.

7. The proposed mechanism relies on the formation of allylic Cu(III) intermediates and the DFT calculation looks fine, however, direct detection of Cu(III) intermediates using EPR spectroscopy

seems to be possible like in *Inorg. Chem.* 2023, 62, 5387–5399. This is not strictly required but authors are encouraged to do so.

[Our response] We appreciate the reviewer's insightful suggestion regarding potential detection of Cu(III) intermediates. After carefully examining the reference (*Inorg. Chem.* 2023, 62, 5387–5399), we note that while Cu(II) species can indeed be detected by EPR spectroscopy due to their paramagnetic nature, Cu(III) complexes would remain EPR-silent because of their diamagnetic character, as correctly pointed out in the cited work.

We would like to clarify that in our current system, we have been unable to isolate the putative Cu(II) and Cu(III) intermediates for direct spectroscopic comparison. This limitation prevents us from obtaining definitive experimental evidence for the proposed Cu(III) species through EPR techniques at this stage.

Reviewer #2 (Remarks to the Author):

The manuscript by Lin and coworkers describes a photocopper-catalyzed system for the synthesis of allylic amines via C–N coupling through outer-sphere nucleophilic substitution at allylic copper(III) complexes. This reaction employs soft nucleophiles, such as alkyl or aryl amines, addressing the issues associated with the traditionally used hard nucleophiles. Through mechanistic studies, the authors experimentally confirmed the radical-mediated formation of allylic copper complexes and provided evidence for the involvement of outer-sphere nucleophilic substitution at allylic copper(III) complexes. This paradigm could be the complement of established methods. I suggest that this manuscript can be very interesting for readers of *Nat. Commun.* and can be accepted after major points below have been addressed.

[Our response]: We appreciate the reviewer's positive feedback and recognition of the potential interest of our work to the readers of *Nature Communications*. We agree with the reviewer's suggestions and have performed additional experiments and analyses to address them. Our detailed responses are outlined below.

1. There are some formatting inconsistencies in the manuscript. For example, in line 101, the nitrogen atom in “N-nucleophiles” is not italicized, and in line 131, “E/Z” in “high E/Z ratios” is not italicized. Please carefully check the manuscript and the Supporting Information (SI) for similar formatting issues.

[Our response]: Thank you very much for your suggestion. The formatting issues have been corrected.

2. The manuscript lacks a reaction condition screening table. It is recommended that the authors extract and summarize key screening data from the Supporting Information (SI) to provide readers with a clearer understanding of the reaction conditions.

[Our response]: Thank you very much for your suggestion. We have extracted some key screening

data into the manuscript. (see Figure 1d)

3. The authors propose a reasonable outer-sphere nucleophilic substitution mechanism at allylic copper(III) complexes. However, based on a survey of recent literature (J. Am. Chem. Soc. 2024, 146, 9444–9454; Angew. Chem. Int. Ed. 2024, 63, e202405560), an alternative pathway could also explain product formation, involving nucleophile coordination to Cu(III) through ligand exchange, followed by reductive elimination. It is recommended that the mechanistic discussion be expanded to address this possibility. Additionally, I believe that the nucleophiles in the system are also likely to pre-coordinate with the copper complex, given the strong basicity of TMG in the system.

[Our response]: Thank you very much for your insightful comments and suggestions. We agree with your assessment. An alternative pathway involving reductive elimination to form the product is plausible for nucleophiles such as carboxylic acids, which have low pKa values. Pre-coordination with the copper complex is also feasible, since the carboxylic acid can be readily deprotonated in the presence of TMG. We have incorporated additional discussion on this point in the manuscript and included supporting DFT calculations in the Supplementary Information.

[Added comments in the manuscript] Nevertheless, alternative mechanistic pathways, including inner-sphere nucleophilic substitution, cannot be ruled out, particularly when employing different nucleophilic partners in three-component reactions. For example, in three-component reaction systems employing TMG (1,1,3,3-tetramethylguanidine) as a base to deprotonate nucleophiles (e.g., carboxylic acids)⁴⁶⁻⁴⁸, the reaction mechanism might shift toward an inner-sphere nucleophilic substitution pathway (Supplementary Figure 15). This alternative mechanism operates stereoselectively, preferentially yielding products with *E*-configuration (Supplementary Figure 16 and 17). Additionally, for the outer-sphere S_N2 reaction proceeding via the allylic copper(III) intermediate pathway with amines as nucleophiles, the thermodynamically favored π-allylcopper complex acts as the key intermediate Cu^{III}- η^3 in Supplementary Figure 18, dictating the selective formation of *E* isomers via a barrier-less route.

4. In the field of copper-catalyzed allylic functionalization, several important studies are worth citing, such as: Angew. Chem. Int. Ed. 2021, 60, 22956–22962; Angew. Chem. Int. Ed. 2024, 63, e202405560; Nature Commun. 2024, 15, 1483.

[Our response]: Thank you for your suggestion and reminding. They have been cited in the Ref 46-48.

Reviewer #3 (Remarks to the Author):

The manuscript of Lin and co-workers reported light-mediated Cu-catalyzed crosscoupling of alkyl iodides and amines (or carboxylic acids) for the synthesis of allylic functionalized

compounds. The presented reactions provide an alternative for Pd catalyzed Tsuji-Trost reactions. Moderate to excellent yields were achieved for the tested substrates and the functional group compatibility is good. Experimental and computational mechanistic studies were performed and an outer-sphere nucleophilic substitution mechanism was proposed accordingly. However, I am not inclined to recommend this manuscript for publication for two main reasons. First, this work is a regular expansion of the authors' previous work on light-mediated Cu-catalyzed coupling of alkyl iodides and amines (ref. 36). Alkenyl-tethered alkyl iodides as substrates or adding 1,3-diene to capture the alkyl radical in situ generated from alkyl iodides were considered in this work, thus allylic functionalized products were obtained. The novelty does meet the criteria of Nat. Commun. Second, the results of mechanistic study were questionable and several issues should be addressed before publication.

[Our response]: We sincerely appreciate your careful evaluation and feedback.

While Ref. 36 focused on direct C-N coupling, the current work achieves allylic functionalization via a distinct radical trapping strategy. Key advances include:

Reactivity Design: Alkenyl-tethered alkyl iodides or added dienes shift the pathway from direct coupling to tandem radical addition/elimination, enabling access to allylic motifs inaccessible via Ref. 36.

Mechanistic Distinction: Our work reveals, for the first time, that copper can facilitate outer-sphere amination at allylic intermediates - a reactivity mode previously observed only with noble metals (e.g., Pd, Ir). This discovery significantly broadens the mechanistic repertoire of copper catalysis, which has been predominantly associated with inner-sphere pathways.

1. The active exciting state for XAT by Cu(I)* complex is singlet or triplet? Computational study or any experiments should be considered to clarify it. The details of the XAT by Cu(I)* complex are important to provide a full mechanistic scheme.

[Our response]: The exciting state for XAT by Cu(I)* complex should be triplet. The detail about the experiments of TA spectra of the copper complex can be found in the Mechanism studies.

The comment in the manuscript:

[Subsequently, we measured transient absorption (TA) spectra over nanosecond to sub-microsecond timescales to investigate the electron transfer process between copper catalyst and alkyl iodide 1. Notably, we observed that the long-lived (2 μ s) ³MLCT state of the Cu(I) complex can be quenched by the corresponding alkyl iodide (1) (Figure 5a,b). The plot of the alkyl iodide concentration against the triplet excited state demonstrates non-linear Stern-Volmer quenching behavior (Figure 5c), which aligns with the inner-sphere electron transfer process involving an exciplex formed between the ³MLCT state Cu(I) complex and the alkyl iodide.³⁶]

2. UV-vis absorption indicated that the mixture of CuI, L1 and TMG is active for light quenching, but TMG is only considered at the last stage just for acid-base neutralization in the computational study. The role of TMG is essentially neglected in the key steps like XAT and C-N coupling.

[Our response] : Thank you for your comments. TMG (as a base) plays a critical role in deprotonating the precursor (L1) to generate the free carbene ligand, which is essential for forming the active copper photocatalyst. This mechanistic step is implicitly accounted for in our computational model by directly using the deprotonated carbene species. Notably, other bases (e.g., K_2CO_3 , Supplementary Table S2) also successfully mediated the reaction, further supporting the role of TMG as a base and the TMG is not essential for the XAT and C-N coupling step. For computational efficiency, the explicit deprotonation step was omitted, as it does not affect the key mechanistic conclusions.

3. Only outer-sphere mechanism was considered for C-N coupling currently, and inner-sphere mechanism should be also calculated for comparison. For example, TMG acts as a base for the deprotonation of amines and carboxylic acids to generate amino anion and carboxylate which further replace the iodide of Cu(II) or Cu(III) complexes followed by reductive elimination of Cu(III) complex to form the C-N or C-O bond.

[Our response] : Thank you for your suggestion. Inner-sphere mechanism, we added the computational analysis for three-component reactions. Please see the Supplementary Figure 15 and Supplementary Figure 16. In that case, the inner-sphere mechanism is possible when using the carboxylic acids as the nucleophiles. We also check the Cu^{III} complexes (**TSB**) for C-N bond formation via inner-sphere pathway in two-component reaction (Supplementary Figure 13).

4. Int_b1 is stable than IntA2 with a formally more electron-deficient Cu center, thus the former seems to be more suitable for the nucleophilic addition of amines where the dissociated iodide could be the base for N-H deprotonation. The outer and inner-sphere mechanisms should be considered from Int_b1.

[Our response] : Thank you for your valuable feedback. After a thorough examination of the possible reaction pathways originating from **Int_b1** by DFT calculation, we found that the outer-sphere S_N2 process proceeds via a barrier-less pathway leading directly to product formation. In contrast, the inner-sphere mechanism requires overcoming a significant energy barrier of 14.4 kcal/mol. Consequently, the outer-sphere S_N2 pathway is energetically more favorable. (Figure 8 and Supplementary Figure 13 &14)

5. The procedure of CV experiments is not clear. How L1-CuI2 complex is in situ generated? How can the authors confirm that the signals in Figure S10 belong to the reduction of L1-CuI2 complex but not other possible complexes? If possible, L1-CuI2 should be synthesized and put to CV experiments.

[Our response]: Thank you for your reminding. We have included detailed descriptions of the CV experiments in the Supplementary Information (Supplementary Figure 10 &11). The L1-CuI₂ was synthesized in situ, with the CV measurements showing a redox potential $E(Cu^{II/I})$ of 0.41 V. For comparison, we also determined the redox potential of in-situ generated L1-CuI, which gave a

similar value of 0.43 V for $E(\text{Cu}^{\text{II/I}})$. Both values are comparable to $E_{\text{calc}}(\text{Cu}^{\text{II/I}})$, further supporting our conclusion that the skipped dienyl radical is unlikely to undergo single-electron oxidation by the Cu(II) complex.

6. The procedure for calculating redox potential in this manuscript is wrong. The term $-(E_{\text{reduced}} - E_{\text{oxidized}})/nF$ is the absolute electrode potential. To get the redox potential relative to AgCl/Ag, the calculated absolute electrode potential should further minus 4.44 and 1.998 eV, respectively. -4.44 eV is the absolute electrode potential of NHE whereas 1.998 is the redox potential of AgCl/Ag relative to NHE. See J. Am. Chem. Soc. 2005, 127, 7227 as an example for redox potential calculation.

[Our response]: Thank you for careful checking the Supplementary information. You are totally right, the $E_{\text{ABS}}(\text{REF})$ should be 4.44 plus 1.998. It is mistake to put 1.998 as the $E_{\text{ABS}}(\text{REF})$ in the previous Supplementary information. However, the calculating redox potential in this manuscript is not wrong, because, actually, we used the corrected $E_{\text{ABS}}(\text{REF})$ to calculate the redox potential. Please see the details as follow.

The redox potential was calculated by the following equation:

$$E_{0/1} = - \left(\frac{E(\text{reduced}) - E(\text{oxidized})}{n_e F} \right) - E_{\text{ABS}}(\text{REF})$$

Where $E(\text{reduced})$ is the energy of neutral system, $E(\text{oxidized})$ is energy of oxidized system, n_e is the number of electrons transferred, F is Faraday constant (96485.33289 C/mol), $E_{\text{ABS}}(\text{REF})$ is the reference potential of Ag/AgCl, 4.639 (4.44V + 0.199V) at 25 °C.

$E(\text{Cu}^{\text{II}}) = -2902.85829945183$ Hatree = -7621454465 J/mol

$E(\text{Cu}^{\text{I}}) = -2903.01123718847$ Hatree = -7621856003 J/mol

Therefore, $E_{\text{calc}}(\text{Cu}^{\text{II/I}}) = 0.48$ vs Ag/AgCl

$E_{\text{rad}} = -234.0727463$ Hatree = -614557995.4 J/mol

$E_{\text{cat}} = -233.8676269$ Hatree = -614019454.4 J/mol

Therefore, $E(\text{rad/cat}) = 0.94$ vs Ag/AgCl

7. It is interesting to see that more acidic phenol and alcohol were inert in the C-N coupling. The competitive C-O couplings were suggested to be considered to check the validity of proposed mechanism.

[Our response]: We appreciate this insightful suggestion. While competitive C-O coupling could theoretically occur, our experimental results show no detectable traces of such byproducts. This strongly indicates that pre-deprotonation is not a prerequisite for the observed C-N coupling selectivity.

8. Is the solvation model used or not in the geometry optimization? The description of computational methods is not clear.

[Our response]: We have added the description in detail.

9. Entropy correction is not suggested, see J. Am. Chem. Soc. 2015, 137, 3811 for the comments on entropy correction.

[Our response]: Entropy correction have been removed.

10. In Figure 1c, a fishhook arrow is missed for the ring-opening of 1,4-skipped dienyl radical.
11. The yields of 3a and 3b should be provided for standard conditions in Figure 1d.
12. In Figure 5a, 'III' should be written in superscript for 94_Int

[Our response]: Thank you for your reminding. All of them have been corrected.

A point-by-point response to the reviewers' comments

Reviewer #1 (Remarks to the Author):

The manuscript has been improved after revision, I recommend the publication of the current manuscript.

[Our response]: Thank you for your valuable comments, which have helped to improve our manuscript. We appreciate your recommendation for acceptance.

Reviewer #2 (Remarks to the Author):

The reviewer considers that the authors have adequately addressed most of the issues, and therefore the manuscript merits publication. This study reports a photocopper-catalyzed system for the synthesis of allylic amines via C–N coupling through outer-sphere nucleophilic substitution at allylic copper(III) complexes, effectively overcoming the limitations associated with conventional hard nucleophiles, and is expected to attract broad attention in the future. I am very pleased to unreservedly recommend the publication of this manuscript in Nature Communications.

[Our response]: We truly appreciate your kind and supportive comments on our manuscript.

Reviewer #3 (Remarks to the Author):

I am still not convinced by the novelty of this paper but I respect the comments of other reviewers and editors. Despite of this, the authors have addressed most of the comments reasonably except the one about solvation effect in geometry optimization. It is stated that “The optimizer implemented in develop version of global reaction route mapping (GRRM) program⁵ were used for all the optimizations of intermediates, transition states (TSS) and intersystem crossing points.” but whether and what solvent model is used for geometry optimization is still unknown. The authors even deleted the description of PCM solvation model in the new SI, does it mean that no solvation model was used for the computational study? That would not be acceptable. Improve to the description of computational details to make the computational results reproducible.

[Our response]: Thank you for pointing this out. As indicated in each figure legend for the computational energy profiles, the solvent effects of *N,N*-dimethylacetamide were accounted for using the PCM model. It was an oversight to omit this detail from the Computational Details section of the Supplementary Information. We have now revised the supplementary material to

include the PCM solvation model description and prevent any potential confusion. We apologize for this oversight and thank you again for ensuring the clarity and reproducibility of our work.

[see the revised computational details] All computations in the present study were performed by Density Functional Theory implemented in Gaussian16c01 program.⁵ The optimizer implemented in develop version of global reaction route mapping (GRRM) program⁶ were used for all the optimizations of intermediates, transition state (TS) and the solvent effects of by the PCM model in *N,N*-dimethylacetamide was used for geometry optimization of all chemical structures. B3LYP functional with the D3 empirical dispersion correction (B3LYP-D3)⁷ was selected as the exchange-correlation functional and Def2TZVP basis set was used for all the atoms. The transition states were verified by frequency calculations and IRC calculations. Gibbs free energy was calculated at 298.15 K.